# Research Progress of Nitrite Metabolism in Fermented Meat Products

**DOI:** 10.3390/foods12071485

**Published:** 2023-04-01

**Authors:** Qiyuan Shen, Xiaoqun Zeng, Lingyu Kong, Xiaoqian Sun, Jingjing Shi, Zhen Wu, Yuxing Guo, Daodong Pan

**Affiliations:** 1State Key Laboratory for Managing Biotic and Chemical Threats to the Quality and Safety of Agro-Products, Ningbo 315211, China; 2Key Laboratory of Animal Protein Food Processing Technology of Zhejiang Province, College of Food and Pharmaceutical Sciences, Ningbo University, Ningbo 315800, China; 3Zhejiang-Malaysia Joint Research Laboratory for Agricultural Product Processing and Nutrition, Ningbo 315800, China; 4School of Food Science and Pharmaceutical Engineering, Nanjing Normal University, Nanjing 210097, China

**Keywords:** nitrogen synthesis, nitrogen degradation, lactic acid bacteria, nitrite reductase, antioxidant, nitrite degradation

## Abstract

Nitrite is a common color and flavor enhancer in fermented meat products, but its secondary amines may transfer to the carcinogen *N*-nitrosamines. This review focuses on the sources, degradation, limitations, and alteration techniques of nitrite. The transition among NO_3_^−^ and NO_2_^−^, NH_4_^+^, and N_2_ constitutes the balance of nitrogen. Exogenous addition is the most common source of nitrite in fermented meat products, but it can also be produced by contamination and endogenous microbial synthesis. While nitrite is degraded by acids, enzymes, and other metabolites produced by lactic acid bacteria (LAB), four nitrite reductase enzymes play a leading role. At a deeper level, nitrite metabolism is primarily regulated by the genes found in these bacteria. By incorporating antioxidants, chromogenic agents, bacteriostats, LAB, or non-thermal plasma sterilization, the amount of nitrite supplied can be decreased, or even eliminated. Finally, the aim of producing low-nitrite fermented meat products is expected to be achieved.

## 1. Introduction

Fermented meat products were developed in response to a demand for meat storage. During the fermentation process, a series of biochemical and physical changes caused by microbial fermentation or enzymes provide the meat products with a unique flavor, color, texture, and antioxidant properties, so as to improve the edible quality of the meat [1]. Subsequently, nitrite, comprising salty white or light yellow particles [2], which primarily contain sodium or potassium nitrite, is commonly used in meat curing [3] as a color protectant, antioxidant, and preservative, inhibiting the growth of spoilage and pathogenic bacteria, such as *Clostridium botulinum* and *Listeria monocytogenes* [4]. In long-term studies, nitrite was investigated as a mammalian vasodilator, which released a protective substance that can save a mammal's life during hypoxia [5]. Nitrite has antibacterial, antioxidant, color development, and flavor production properties, making it almost irreplaceable [3]. Therefore, nitrite is commonly utilized in the processing of meat products.

Regulation No.1333/2008, which applies to sausages, pig hoofs, and other products in the European Union (EU), previously regulated the use of nitrite to a maximum dosage of 150 mg/kg [6]. Excessive nitrite might pose a threat to food safety. Firstly, once nitrite enters the body, it causes hypoxic poisoning by binding to hemoglobin in the blood. Secondly, after ingestion of animal products, nitrite reacts with secondary amines to form nitrosamine [7]. Increasing awareness of health care advances meant that people gradually realized the harm of excessive nitrite to the body, causing concerns regarding fermented meat product ingredients such as nitrite, which has had a massive effect on cured meat product sales. Therefore, preferences for the consumption of safe, low-salt, and low-fat meat products without chemical additives have increased considerably [8]. Several investigations have been undertaken to determine whether antioxidants, chromogenic compounds, and bacteriostatic agents can substitute for nitrite in the curing process [9]. However, cured meat products are complex products, in which each ingredient plays a special role, making reformulation difficult. Challenges arise as a result of the reduction of salt and chemicals in cured meat products, which alter their sensory qualities and might have an impact on the microbial ecosystem, resulting in uncharacteristic and unsafe products [8].

Recent research reviewed the role of nitrite in fermented meat products, and numerous nitrite metabolic pathways have been documented and collated. The LAB in fermented meat products are the key factors affecting nitrite metabolism, and the acids, enzymes, and other compounds produced by LAB contribute to nitrite synthesis and degradation. Consequently, this review summarizes research from the previous years, the majority of which were published within the last five years, and discusses nitrite’s source, degradation pathway, and conversion regulation from the four perspectives of nitrogen synthesis and degradation, nitrite metabolism, nitrite safety, and methods and application of fermented meat products. This review mainly describes the role of LAB in the degradation of nitrite in fermented meat products to provide a theoretical foundation for future research into nitrite and its applications.

## 2. Microorganisms Related to Nitrite Metabolism

Nitrite is one of the nitrogen cycle intermediates on Earth [10]. Nitrogen in nature is mainly present as NH_4_^+^, as the inert N_2_, the lowest oxidized state, and as NO_3_^−^, the highest oxidized state. The transition between NO_3_^−^ and NO_2_^−^, NH_4_^+^, and N_2_ in the biogeochemical cycle constitutes the balance of nitrogen synthesis and degradation through nitrate and denitrification bacteria (Figure 1).

Nitrifying bacteria are a class of aerobic bacteria with two physiological subgroups, *Nitrobacter* and *Nitrosomonas*. *Nitrobacter* (also known as nitrite-oxidizing bacteria), containing nitrite oxidase, oxidize nitrite to nitrate (reaction (1)) and reproduce for a generation in 18 h. *Nitrosomonas* (nitrite bacteria, also known as ammonia-oxidizing bacteria), oxidize NH_4_^+^ to nitrite (reaction (2)) and reproduce for a generation in 18 min [11]. The growth and propagation rates of *Nitrobacter* are significantly lower than those of *Nitrosomonas*. The conversion rate of ammonium nitrogen to nitroso nitrogen is significantly slower than that of nitrate-nitrogen to nitrite-nitrogen, leading to the accumulation of nitrite-nitrogen.
2NO_2_^−^ + O_2_ → 2NO_3_^−^ + Energy (1)
2NH_4_^+^ + 3O_2_ → 2NO_2_^−^ + 4H^+^ + 2H_2_O + Energy (2)

Denitrifying bacteria are a group of bacteria that reduce nitrate-nitrogen to N_2_ and NH_4_^+^. Mostly heterotrophic, facultative anaerobic bacteria, such as *Bacillus stephensi* and *Trichomonas aeruginosa*, contain a variety of nitrite reductase (NiR) enzymes to degrade nitrite. Copper-type nitrite reductases (CuNiRs) and cytochrome cd1 nitrite reductases (cd1NiRs) convert NO_3_^−^ to N_2_ (reactions (3) and (4)) by denitrification, and polyheme c nitrite reductases (ccNiRs) convert NO_3_^−^ to NH_4_^+^ by amination. Denitrifying bacteria maintain the nitrogenous nitrogen content at a steady low-concentration level and the nitrogen cycle remains in dynamic equilibrium [12].
C_6_H_12_O_6_ + 12NO_3_^−^ → 6H_2_O + 6CO_2_ + 12NO_2_^−^ + Energy (3)
5CH_3_COOH + 8NO_3_^−^ → 6H_2_O + 10CO_2_ + 4N_2_ + 8OH^−^ + Energy (4)

Bacteria with degradation effects mainly include *Lactiplantibacillus plantarum*, *Levilactobacillus brevis*, *Leumesenteroides*, *Pediococcus cerevisiae*, *Streptococcus faecalis*, and others. Paik et al. found that *Lactiplantibacillus plantarum* KGR5105, *Levilactobacillus brevis* KGR3111, *Latilactobacillus curvatus* KGR2103, and *Lactobacillus serans* KGR4108 also produce NiR. LAB showed significant degradation ability in fermented meat products under optimal conditions [13].

Fermented meat products, which typically contain nitrite, a nitrogen-containing dietary additive, follow the same natural rhythm of nitrogen synthesis and breakdown. In general, the pH of meat products ranges between 5.5 and 6.2, while the pH of dry fermented sausage ranges between 4.5 and 5.5 [14], both of which are optimum conditions (pH 4.5–5.5) that allow LAB to grow. However, *N*-nitrosamine formation may be more easily achieved in dry fermented sausages as the pH of the product approaches the optimum pH (pH 3.5) of the nitrosation reaction [14,15]. Nitrite is synthesized and degraded by bacteria in the endogenous system of fermented meat products, completing the nitrite metabolism cycle in these items [10].

## 3. The Role of Nitrite in Fermented Meat Products

### 3.1. Color Formation Effect

NO reacts with myoglobin (Fe^2+^) and methemoglobin (Fe^3+^) in fermented meat products to form cured pink, which serves as a coloring and antioxidant. Both free and heme-bound iron is the principal pro-oxidant in meat products [16]. NO binds to myoglobin (Fe^2+^) to form unstable NO-myoglobin, which is converted to a stable pink pigment, nitroso-heme, upon heating and prevents iron-induced oxidation [17]. Myoglobin can also react with nitrite and be oxidized to methemoglobin (Fe^3+^), which in turn reacts with NO to form NO-methemoglobin, and reducing agents restore NO-methemoglobin to form NO-myoglobin and finally form cured pink under heating [18]. Postmortem, active cytochrome enzymes possess the ability to utilize oxygen, which is responsible for the red color on the surface of meat caused by the presence of oxymyoglobin [19]. Similarly, nitric oxide promotes the lipid peroxidation cycle due to its lipophilic nature by reacting with alkyl-, alkoxyl-, and peroxyl radicals [20]. Consequently, nitrite can inhibit both primary and secondary oxidation.

### 3.2. Flavor Improvement

The exact mechanism of the effect of nitrite on flavor is still unclear, but the antioxidant activity of nitrite is an important factor affecting its flavor, although other antioxidants added to meat products cannot show the unique flavor of fermented meat products. Nitrite will not directly produce a specific flavor substance in fermented meat products. However, nitrite can inhibit lipid oxidation, which inhibits the production of aldehydes, such as hexanal and pentanal, which masks the sulfur-containing compounds that make meat products produce a pickled flavor [21]. On the other hand, nitrite can induce the formation of Strecker aldehyde, which is formed via the degradation of amino acids by dicarbonyls that are produced from the Maillard reaction, due to the pro-oxidant effect of nitrite, and is related to food flavor [22].

### 3.3. Antioxidant Properties

Nitrite can also be used as an antioxidant by receiving oxygen from sensitive molecules or producing active nitrogen. Villaverde et al. reported that the formation of carbon-based compounds in fermented sausage increased with the increase in nitrite content [21]. Vossen and De Smet studied the effect of sodium nitrite on protein oxidation in isolated myofibrillar protein and porcine patties [23]. It was found that the TBARS value of sodium nitrite was significantly lower than that of the control group (without sodium nitrite) but had no effect on the content of carbon-based compounds as an indicator of protein oxidation. In addition, when sodium nitrite and sodium ascorbate are used together, the yield of carbon-based compounds will increase, but it will not increase when used alone, which is the result of nitrite as an oxidant of ascorbate. Dehydroascorbate is produced by the oxidation of ascorbate by nitrite, which is similar to the carbon group of reducing sugar. It produces non-protein carbon-based compounds via non-enzymatic glycosylation [24].

### 3.4. Antimicrobial Effect

Nitrite is the substance of interest in microbial inhibition, but it is not the nitrite itself that produces the inhibition, which is closely related to its ability to trigger the formation of NO. In vitro, NO can directly react with microbial proteins that contain an iron–sulfur enzyme cluster (Fe-S-NO) and form protein-bound dinitrosyl dithiolato iron complexes [25]. Ferredoxin in *Clostridium* spp. is involved in ATP synthesis by the microorganisms in this Fe-S broad group. Therefore, this Fe-S-NO mechanism is considered to be the main factor inhibiting the growth of *Clostridium* spp. vegetative cells, and it can also be observed in aerobic and facultative pathogens associated with cured meat products [26]. The dinitric iron complex formed by NO binding to proteins is present in prokaryotic and eukaryotic cells, and iron–sulfur proteins are the major source of protein-bound dinitrosyl iron complexes formed in *Escherichia coli* cells under nitric oxide stress [27]. At the same time, peroxynitrite, a highly oxidized and unstable compound, may be involved in the inhibition mechanism of NO, and the formation of peroxynitrite is related to the changes in the oxidation state of intermediate compounds during the reduction of myoglobin and nitrate. Due to its oxidation and nitrification ability, it will damage proteins, DNA, and lipids, and ultimately inhibit the growth of microorganisms [28].

## 4. Metabolic Pathways of Nitrite in Fermented Meat Products

Exogenous manufacture, contamination during processing, and endogenous microbial synthesis are the main sources of nitrite in fermented meat products. In addition, a small amount of nitrite is produced by contamination during rearing and endogenous microbial production. Acids, enzymes, and non-acid and non-enzyme compounds in bacteria can degrade nitrite in fermented meat products (Figure 2).

### 4.1. Source of Nitrite in Fermented Meat Products

#### 4.1.1. Exogenous Manufacture during Processing

The addition of nitrite to meat products during processing improves their appearance, color, flavor, and safety. It is also the main source of nitrite in fermented meat products. In meat products, nitrite serves four purposes. The first is the reaction of nitric oxide (NO) with hemoglobin and myoglobin to form nitrohemoglobin and nitrosymyoglobin, which maintain the bright color of meat products [29]. Jarulertwattana found that 4 × 10^−6^ g/g nitrite was suitable for curing chicken when used with ginger sauce, but more than 4 × 10^−6^ g/g nitrite will cause pink defects in chicken legs [30]. Second, nitrite can inhibit the growth of *Clostridium botulinum* and *Staphylococcus aureus*. The minimum concentration of nitrite that inhibits the outgrowth of Clostridium botulinum is from 4 × 10^−5^ to 8 × 10^−5^ g/g [1]. Third, nitrite has an antioxidant effect and can suppress lipid peroxidation, lowering the production of foul odors. Ji found that sodium nitrite can effectively inhibit lipid oxidation at 1 × 10^−4^ g/g during the curing process of mutton [31]. Finally, nitrite can aid some fermentation bacteria in the production of fermented flavor components in cured meat products through metabolism or the hydrolysis of protease and lipase. [4,32].

Some vegetable extracts are added to fermented meat products as nitrite substitutes because they contain a large amount of nitrite: celery, watercress, lettuce, spinach, and rape (<2500 mg/100 g); Chinese cabbage, leek, and parsley (1000 to <2500 mg/100 g); radish and cabbage (500 to <1000 mg/100 g); carrots, cucumbers, pumpkins, and broccoli (200 to <500 mg/100 g); and potatoes, tomatoes, onions, eggplants, mushrooms, and asparagus (<200 mg/100 g) [33,34]. Meat products cured with vegetable extracts showed similar quality and sensory properties to nitrite-cured products [35]. Natural nitrite sources have a higher consumer preference than synthetic nitrite because they contain a variety of antioxidants and antibacterial compounds. Phenolic compounds in vegetable extracts may hinder the conversion of nitrite into *N*-nitrosamines, making the probability of nitrosamines production lower than that of synthetic nitrite. However, there is no relevant research to prove its conversion relationship. Nevertheless, these plant extracts have many application limitations. Firstly, only vegetables that contain enough nitrate can be used. Furthermore, the taste and color of vegetables will also affect the sensory quality of fermented meat products. Furthermore, the transformation of nitrate to nitrite in plant extracts is uncontrollable, and the content of nitrite finally transformed may not reach the desired level [36].

#### 4.1.2. Contamination during Rearing

During animal rearing, domestic animals will ingest nitrite in the environment. Nitrogen compounds are widely used as fertilizers in agriculture. Crops produce plant chlorophyll in response to nitrogen fertilizer, which increases crop yield [37]. The use of nitrogen fertilizer has risen substantially in recent years, particularly in Southeast Asia [38]. Despite having a positive and significant impact on global food production, nitrogen fertilizer has a significant negative impact on the larger ecosystem. Under natural conditions, nitrifying microorganisms in the environment influence the nitrogen cycle of water, ensuring that nitrogen metabolism remains balanced. However, a large amount of high-nitrogen metabolites are added to aquaculture water, which outperforms the metabolism of natural bacteria in the reservoir, alters the dynamic balance of the nitrogen cycle, causes nitrite to deposit in the water, and eventually accumulates nitrite in animals [39].

Unintentionally added nitrate and nitrite in livestock products have been reported. Iammarino et al. reported a maximum endogenous nitrate content of 30 mg/kg in horse meat and beef, and up to 40 mg/kg in fresh pork meat [40]. Iacumin et al. also determined that the thresholds for nitrite and nitrate concentrations in raw meat were less than 4 mg/kg and 22 mg/kg, respectively, which are below the threshold; the authors believed that nitrite and nitrate in raw meat were not intentionally added [41].

#### 4.1.3. Endogenous Microbial Production

In addition to the nitrite accumulated by exogenous addition and contamination of raw materials, a tiny quantity of nitrite is produced during meat product fermentation by endogenous microbial action. In ammonia-oxidizing bacteria, monoamine oxidase converts NH_4_^+^ to hydroxylamines, which are then converted to nitrite by hydroxylamine oxidase (reaction (2)). Denitrifying bacteria can also convert nitrate-nitrogen to N_2_ and NH_4_^+^, with nitrite as an intermediary product. Iacumin et al. investigated dry pickled ham from San Daniele without adding nitrite for 14–19 months and found that nitrite in the ham was present at less than 4 mg/kg [42]. When products have a long curing process, adding nitrate instead of nitrite to meat products, the nitrate is reduced to nitrite by microbial nitrate reductases [43].

### 4.2. The Degradation Pathway of Nitrite in Fermented Meat Products

LAB and denitrifying bacteria degrade nitrite in fermented meat products, mostly through acid and enzyme degradation, while other non-acid and non-enzyme compounds can also degrade nitrite. LAB can also limit the metabolism of nitrate reduction bacteria, reducing the generation of nitrite, and inhibiting the growth of *Escherichia coli*, *Klebsiella pneumoniae*, *Pseudomonas fluorescens*, *Pseudomonas alkalogenes*, and other bacteria [44]. Some bacteria, such as *Latilactobacillus sakei*, *Latilactobacillus curvatus*, and *Levilactobacillus brevis*, have a strong ability to reduce nitrite by inhibiting the formation of *N*-nitro-sodimethylamine precursors [45]. Wu et al. used a co-culture of *Lactiplantibacillus plantarum* Shanghai brewing 1.08 and *Zygosaccharomyces rouxii* CGMCC 3791 to minimize nitrite and biogenic amine concentrations and increase the flavor components in Chinese sauerkraut [46].

The degradation by LAB is separated into two stages according to Zeng et al. [47]. In the early stage of fermentation, when the pH > 4.5, nitrite is mostly degraded enzymatically. Second, because LAB create acid, the breakdown of nitrite in late fermentation is primarily acid degradation after pH < 4.5.

In the curing process, LAB mostly focused in the processing step degrade nitrite rather than store it. According to Huang, *Limosilactobacillus fermentum* RC4 and *Lactiplantibacillus plantarum* B6 were added to cured meat. The content of nitrite was significantly reduced (0.75 mg/kg) during processing, which was significantly lower than that of regular cured meat (4.67 mg/kg); during 0–20 days of storage, the moisture content of bacon decreased significantly, the pH increased continuously, but the content of nitrite did not change significantly [48].

According to reports, nitrite is degraded in the following percentages in meat products: reactions with heme proteins (5–15%), non-heme proteins (20–30%), nitrous acid with free-amino acids via the Van Slyke reaction to generate nitrogen gas (1–5%), sulfhydryl groups (5–15%), lipids (1–5%), oxidation back to nitrate (1–10%), and free nitrite (5–20%) [49].

#### 4.2.1. Acid Degradation

The acid degradation of nitrite demonstrates that nitrite serves as a colorant, antioxidant, unique flavoring agent, oxidant, and bacteriostatic agent in fermented meat products. Nitrite also can produce *N*-nitrosamines with a carcinogenic teratogenic effect.

Fermented meat products contain large amounts of H^+^ after microbial fermentation, and nitrite and H^+^ are decomposed into NO and H_2_O (reactions (5)~(7)) [4].
NO_2_^−^ + H^+^ ↔ HNO_2_
(5)
2HNO_2_ ↔ N_2_O_3_ + H_2_O (6)
N_2_O_3_ ↔ NO + NO_2_
(7)

Nitrite (NO_2_^−^) forms nitrite acid (HNO_2_) when it combines with hydrogen ions (H^+^). Nitric acid then gradually decomposes into nitrous trioxide (N_2_O_3_) and water molecules (H_2_O) (reactions (6) and (7)). N_2_O_3_ further dissociates into NO and NO_2_ (reaction (8)). NO can further form N-nitrosamines with carcinogenic and teratogenic effects [3]. In the endogenous environment, the intermediate between nitrite and NO, NO_2_, establishes a dynamic balance.

#### 4.2.2. Enzyme Degradation

Nitrite is degraded by NiR and reduced to NO or NH_3_. NiR, which is a critical enzyme in the nitrogen cycle, catalyzes nitrite reduction and mainly comes from LAB and denitrifying bacteria. The majority of NiRs are intracellular enzymes that are mostly found in the periplasmic region or cell membrane, and some are free in the cytoplasm. Liu et al. reported that NiR produced by *Lacticaseibacillus rhamnosus* LCR6013 might, via the nitrate respiration pathway (NO_2_ → NO → N_2_O → N_2_), produce nitrous oxide (N_2_O) and degrade nitrite [50].

Depending on the reactants and cofactors, NiRs can be classified into CuNiRs, cd1NiRs, ccNiRs, and ferredoxin-dependent nitrite reductases (FdNiRs) [51]. The genotypes of NiR mainly include *nir*K, *nir*S, *nrf*A, and *nir*B, etc., of which *nir*K and *nir*S are key genes in the denitrification process, encoding a CuNiR and cd1NiR, respectively [52]. *nrf*A and *nir*B are the key genes of nitrate ammonification, which reduce nitrite by transferring six electrons. Genes *nrf*A and *nrf*H encode the double-subunit complex ccNiR protein composed of two subunits, NrfA and NrfH, respectively.

CuNiRs are involved in the denitrification of nitrogen during metabolism. The reduction of NO_2_^−^ by CuNiRs can be divided into five steps: the combination of NO_2_^−^ with the enzyme, the reduction reaction, dehydration of bound intermediate products, NO release, and enzyme regeneration. NO_2_^−^ combines with the T2Cu center in the oxidized form to replace a soluble molecule and forms a hydrogen bond between the Asp98 residue and an oxygen atom of NO_2_^−^. When the electron is transferred from T1Cu to T2Cu, the proton of the hydrogen bond is transferred from the Asp98 residue to the oxygen atom of the substrate to form an intermediate product O=N-O-H. The N-O bond of the oxygen atom is then broken and the product NO is released into the active center [53].

Similar to ccNiRs, the NiRB large and NiRD small subunits are encoded by the genes *nir*B and *nir*D, respectively, and are used to degrade nitrite by forming the NiRBD complex [54]. NiRB encoded by the *nir*B gene, catalyzes NO_2_^−^ to NH_4_^+^, and is a soluble NADH-dependent NiR catalytic subunit-containing sirol heme. NiRB, under anaerobic conditions, uses NADH as an electron donor to reduce nitrite in the cytoplasm, while oxygen suppresses its activity [55]. Wang et al. used anaerobic chemostat culture technology, in which NiRB-lacZ was used to report fusion steady-state gene expression, revealing the differential expression pattern of gene *nir* in *E. coli* [56]. It was found that NiRB is an inducible enzyme, and its expression is induced by high nitrate and nitrite conditions. The synthesis of *nir*B-encoded NiRB is high only when the content of NiRB exceeds the consumption capacity of the cells.

Microorganisms can spontaneously regulate the content of nitrite in fermented meat products, which is mainly related to the genes in the microorganisms. The exogenous environment, the normal expression of genes, the deletion of gene fragments, and the interaction between multiple genes will affect the regulation of nitrite metabolism in microorganisms [57]. Zeng et al. used proteomic and bioinformatic analyses to identify 31, 87, and 190 differentially expressed genes in the process of nitrite degradation in *Limosilactobacillus fermentum* RC4, including *adh*E and *lpd*A which are involved in carbohydrate metabolism, *cys*K related to amino acid metabolism, *nir*B corresponding to nitrogen metabolism, *fab*I and *acc*D associated with lipid metabolism, and *gsk* involved in nucleotide metabolism [47]. These genes are involved in the metabolism of *Limosilactobacillus fermentum* RC4 during nitrite reduction. Chu et al. studied the effects of carbon sources (acetate and glucose) on the endogenous denitrification and ammoniation of *Candida* [58]. After adjusting the oxygen–phosphorus ratio, it was found that acetate (54.2%) had a higher efficiency of converting nitric acid into nitrite (90.2%), whereas glucose (51.3%) made the accumulation of nitrite more stable (85.3%). The total nitrogen removal efficiencies of acetate (88.8%) and glucose (91.3%) were similar (87.8% and 89.8%). Iino et al. studied the genes *ro06366* and *ro00862* of *Rhodococcus* rHA1 [59]. The single mutant with deletion of the above gene showed growth retardation in the environment when using nitrate or nitrite as the only nitrogen source. Iino allowed the double mutant to grow in the environment with a nitrate and nitrite nitrogen source. It was found that nitrate and nitrite were not the only nitrogen sources used and both *ro06366* and *ro00862* are involved in the regulation of nitrite and nitrate. Khlebodarova et al. studied the process of nitrite utilization by NiR in *E. coli* cells [60]. NO_2_^−^ is reduced to NH_4_^+^ outside the cell via NrfA reductase, flows into the cell through the NirC transporter, participates in the degradation of proteins and their complexes, and is converted to NH_3_ through NirB reductase, and then flows out of the cell through the NirC transporter.

#### 4.2.3. Other Substances for Degradation

In addition to the aforementioned degradation pathways, several non-acid and non-enzyme substances, such as flavonoids, polyphenols, and ascorbate, also have effects in nitrite scavenging. Guo et al. measured the nitrite scavenging ability of flavonoids using the diazotization-coupling reaction in vitro, and found that the nitrite scavenging activity was closely related to phenolics (r^2^ = 0.990, *p*  <  0.01) and flavonoids (r^2^ = 0.923, *p*  <  0.05) [61]. Ben et al. found that polyphenols in sea buckthorn had a higher nitrite-degrading capacity than other compounds, with a nitrite-degrading rate of 75.9% [62]. Skibsted found that antioxidants such as ascorbate and polyphenols can induce the reduction of N_2_O_3_, promote the production of NO, and destroy the balance between nitrite, NO, and NO_2_, ultimately reducing the content of nitrite [20]. Feng et al. reported the effect of nitrite on protein oxidation and nitrification of cooked sausage, in which the antioxidant effect of nitrite on protein oxidation was manifested as a significantly reduced base content, higher free amines, and lower surface hydrophobicity [63].

### 4.3. Factors Affecting Nitrite Degradation

#### 4.3.1. Effects of the Food Matrix on the Degradation of Nitrite

Polysaccharides and inorganic salts in the food matrix mainly affect the degradation of nitrite. The food matrix influences the activity of catalase in vitro of LAB and produces hydrogen peroxide, altering the ability of autolysis in vitro of LAB and influencing the activity of nitrate reductase in vitro [64]. Seo et al. studied the extracellular polysaccharides produced by *Lactiplantibacillus plantarum* YML009 [65]. The nitrite clearance rate was 43.93%, which proved the extracellular polysaccharides have a strong ability to degrade nitrite, and the higher the sugar content, the higher the nitrite-free radical clearance rate. Inorganic salt can reduce the water content of fermented meat products and inhibit the growth of microorganisms. When excessive sodium chloride (NaCl) is used to pickle fermented meat products, bacteria will dehydrate and die, and the nitrite degradation effect will be significantly reduced. Delgado-Pando et al. evaluated bacon with different salt contents and found that the color of bacon with a high salt content was redder than that with a low salt content [66].

#### 4.3.2. Effects of Fermentation Conditions on the Degradation of Nitrite

The lower the pH, the more appropriate the fermentation temperature, and the longer the fermentation duration, which leads to less nitrite production. Kilic et al. investigated the influence of pH on residual nitrite in cured meat products and discovered that lowering the pH greatly reduced residual nitrite, whereas increasing the pH increased residual nitrite [67]. Under an acidic environment, the microbial system in fermented meat products is regulated spontaneously, and *Lactobacillus acidophilic* has competitive advantages to inhibit other bacteria and possible pathogens [68]. Furthermore, the longer the fermentation time is, the more dominant the LAB will be.

## 5. Safety of Nitrite in Food

### 5.1. Toxicity of Nitrite

NO is a crucial compound generated from nitrite, which combines with myoglobin and methemoglobin, proteins in cells, and with amines to lower NO levels in the dynamic balance, resulting in nitrite degradation. In meat fermentation, NO has coloring, antioxidant, cured flavor, and antibacterial benefits; however, it can also be carcinogenic. At low concentrations (nm), NO acts as a signaling molecule for intercellular communication in neurons and the cardiovascular system [69]. At high concentrations (μM), NO can kill pathogens and cancer cells [70].

Through a nucleophilic substitution reaction, NO can combine with non-protonated secondary amines to form *N*-nitrosamines, which are carcinogenic and teratogenic [3]. The most common volatile *N*-nitrosamines in meat products are *N*-nitrosodimethylamine, *N*-nitrosodiethylamine, *N*-nitrosopiperidine, *N*-nitrosopyrrolidine, and *N*-nitrobenzylmorpholine. Among them, in terms of oncogenicity and genotoxicity, *N*-nitrosodimethylamine and *N*-nitrosodiethylamine are regarded as the most volatile *N*-nitrosamines [71]. The amount of *N*-nitrosamines production mainly depends on the intake of nitrite and the processing conditions in meat products, which also increases in concentration under conditions of pH 2.5~3.5, longer storage time, and high-temperature and high-acid conditions [72]. The presence of amines is one of the primary prerequisites for the creation of *N*-nitrosamines. There are not many *N*-nitrosamines in raw meat since amino acids are only decarboxylated to produce amine. Nevertheless, procedures including maturation, fermentation, and curing may increase their synthesis of *N*-nitrosamines. It was discovered during processing that the quantity of precursors was directly correlated with the amount of *N*-nitrosamines in meat products, but even if precursors were present, low water activity and an unfavorable pH would prevent the production of nitrosamines in meat products [73]. Additionally, several cooking techniques that reach temperatures greater than 130 °C, such as frying or barbecuing, may make it more likely that *N*-nitrosamines will occur [3]. To prevent nitrosation, chemicals such as ascorbate and tocopherol are added during processing. The presence of sulfhydryl compounds, some phenols, and tannins in meat products might also inhibit the formation of *N*-nitrosamines [15].

### 5.2. Addition Limit of Nitrite in Fermented Meat Products in Different Countries

According to current knowledge, the use of nitrite mainly considers two risks—the final formation of *N*-nitrosamines and the final growth of serious pathogens. Nitrite has potential carcinogenicity in the human body, and its intake should be limited [7]. In addition to the processing steps of curing meat products, consumers’ cooking methods with temperatures higher than 130 °C, such as frying or grilling meat products, will also increase the possibility of *N*-nitrosamines formation [3]. By contrast, other studies have shown that nitrite, as a vasodilator in mammals, is a life-saving medication that releases a protective substance during hypoxic events [5]. The European Food Safety Authority (EFSA) assessed the acceptable daily intake (ADI) in 2017, excluding infants under 3 months of age, and determined that the ADI of nitrate is 3.7 mg/kg body weight (bw)/d and that of nitrite is 0.07 mg/kg bw/d. Most countries’ management regulations allow for the use of nitrite (Table 1).

### 5.3. Clean Label Movement

In some countries, there are plant extracts such as pre-converted nitrites added to fermented meat products, and these products are labeled “natural”, “organic”, and “uncured” [78], which may lead to confusion or even mislead consumers. In addition, the vegetable extract’s precise chemical makeup need not be disclosed, which also encourages some businesses to use it in place of some chemical additions [8]. On the other hand, the use of plant extracts rich in nitrite does not avoid the production of *N*-nitrosamines because the residual nitrite in the product probably reacts under high-temperature conditions and may produce *N*-nitrosamines [79]. According to the Standing Committee on Plants, Animals, Food, and Feed (ScoPAFF), the use of vegetable extracts with a high nitrite content is regarded as an addition in the food processing process, and regulation N°1333/2008 also ensures the use of vegetable extracts in food processing [80]. Therefore, adding vegetable extracts to fermented meat products as nitrite to label items green is a false tactic used by food producers and operators [81]. Rivera et al. thought that the elimination of the “uncured” labeling policy for meats processed with pre-converted nitrites from vegetable sources would improve transparency for consumers [1].

## 6. Methods and Applications of Substituting Nitrite in Fermented Meat Products

Nitrite can inhibit the growth of bacteria due to its oxidation and nitrification ability. The survival or multiplication of microorganisms under reduced or in the absence of nitrite concentrations is increased substantially and represents a challenge for the meat industry to guarantee the safety of fermented meat products. Commercially speaking, it may provide value to draw customers if nitrite is not listed on the label of fermented meat products.

Researchers continue to find safe substitutes for nitrite, including plant substitutes, microbial substitutes, and organic acid substitutes. However, at present, no substitute can currently match the entire action of nitrite and completely replace it. Therefore, the most successful option is to use low-dose nitrite in the curing process of meat products in combination with other substances or technologies to provide anti-corrosion, bacteriostasis, color, and taste, while also preserving the quality of meat products as much as possible.

Recently, many studies have been conducted on the inhibition nitrite synthesis (Table 2). It is required to prevent the synthesis of nitrite in order to lower the level of nitrite in fermented products, such as by adding antioxidants and LAB, and replacing the role of nitrite with chromogenic agents and antibacterial agents.

### 6.1. Antioxidant Substitutes

As an antioxidant, nitrite can largely influence flavor to prevent lipid oxidation and the formation of undesirable lipid oxidation byproducts such as hexanal and 2,4-decadienal [21]. Its antioxidant substitutes can be divided into synthetic antioxidants and natural antioxidants. Synthetic antioxidants include propyl gallate, tert butyl hydroquinone, and butyl hydroxyanisole. Natural antioxidants include phytic acid, vitamin E mixed concentration, guaiac resin, flavonoids, amino acids, and others [90]. Some natural ingredients not only have an antioxidant effect, but also have a potential inhibitory effect on controlling microbiological hazards, such as cloves, essential oils of aromatic and medicinal plants, plant extracts with high polyphenols concentration, acidified whey, honey, and other bee products [34].

### 6.2. LAB and Its Enzymes

LAB are the ideal bacteria in food processing. The vast majority of LAB have a status generally recognized as safe and they have good salt resistance. In addition, the European Food Safety Authority has granted the status of qualified presumption of safety to many LAB, included in the genera *Carnobacterium*, *Lactococcus*, *Leuconostoc*, *Oenococcus*, *Pediococcus*, *Streptococcus*, and the former *Lactobacillus* genus, recently reclassified into twenty-five new genera [91,92]. In the environment of pickled food, LAB can ferment food and increase flavor, and also produce acetic acid, bacterial peptides, and hydrogen peroxide to inhibit harmful bacteria [93,94]. LAB contain special enzymes that can reduce nitrosamine, and the lactate can also reduce nitrite content [44]. When LAB and other substitutes are used at the same time, it is very important to select the most appropriate combination, so as to produce favorable results and even synergy.

### 6.3. Chromogenic Agent Substitutes

Nitrite has a chromogenic effect in the pickling process. The natural pigments that can replace nitrite include *monascus* red, nitrosohemoglobin pigment, chili red, beet red, sorghum red, carboxyhemoglobin, and tomato red. The replacement synthetic pigments include carmine, erythritol, sunset yellow, lemon yellow, and amaranth. Some plant extracts, such as beetroot and berries extracts, can be used to impart the red color of nitrite-free meat products to develop and stabilize color [85]. However, the color produced by chromogenic agent substitutes is frequently different from that produced by traditional meat products. Patarata et al. reduced the nitrite content of cured loins made from wine- and water-based marinade, and studied the consumer’s evaluation of product color [95]. Consumers prefer nitrite-free products even though the color of fermented meat products created with color additive alternatives is subpar if the label says “additive-free”.

### 6.4. Bacteriostat Substitutes

Nitrite can effectively prevent food spoilage and inhibit the growth of *Clostridium botulinum* and other biological hazards, including *Listeria monocytogenes*, *bacillus cereus*, and *Staphylococcus aureus* [96,97]. The antipathogenic properties of nitrite are mainly attributed to three aspects. The first aspect is perturbing oxygen uptake and oxidative phosphorylation; the second is inhibiting critical enzymes, including aldolase, glyceraldehyde-3-phosphate, and nitrogenases; and the third is forming bactericidal nitrite derivatives [98]. Substitute bacteriostatic agents for nitrite include organic acids, tea polyphenols, spice extract, and bacteriocin [99]. On the other hand, some physical methods of food preservation can be mobilized to control microbial hazards, namely, the use of high isostatic pressures, activated plasma, pulse-field UV light, and active packaging [100,101,102].

However, these bacteriostat substitutes show several limitations. Firstly, their ability to inhibit microbial growth may be lower than nitrite. Secondly, some of these natural antimicrobials and plasma-treated water have a selective effect on Gram-positive or Gram-negative pathogens, vegetative cells, or spores. Thirdly, the components of some vegetable extracts will greatly change the aroma and color of products, which affects their application in the production and processing process [1].

### 6.5. Non-Thermal Plasma Sterilization

Some emerging food sterilization methods, such as non-thermal plasma sterilization, can be used for the preparation of fermented meat products without nitrite [30]. The interaction of plasma with water can result in the generation of nitrate and nitrite, as well as reactive oxygen species. Reactive oxygen and nitrogen compounds react with cellular macromolecules, such as proteins, lipids, enzymes, and DNA, to change the functional features of biofilms, affecting normal physiological functions, and ultimately leading to microbial cell death [96]. However, during non-thermal plasma sterilization, oxygen atoms react with vibration-excited nitrogen molecules to produce nitric oxide (NO) (reaction (8)).
O + N_2_→NO + N(8)

When NO combines with plasma, it forms nitrogen oxide (NO_2_) and additional nitrogen oxides (NO_3_, N_2_O, N_2_O_3_, and N_2_O_5_) [103]. Nitrogen oxides diffuse and dissolve in liquids, where they react with water molecules to create nitrite, nitric acid, and nitrate acid, followed by nitrite acid degradation [36].

After non-thermal plasma sterilization, Yong et al. marinated tenderloin in brine without sodium nitrite and observed that the color was similar to that of sodium nitrite-treated tenderloin, with an increased red a* and no significant difference in yellow b* or brightness L* [104]. Lee et al. used plasma for 30 min to produce pork batter with 42.42 mg/kg of nitrite, with no significant differences in physicochemical or sensory performance when compared with pork batter prepared using sodium nitrite or celery powder [105].

## 7. Summary

Nitrite is a multifunctional food additive that is widely used in the fermentation of meat products. However, high levels of nitrite residues have become a significant concern that impacts the safety of fermented meat products and restricts the growth of the traditional fermented meat product industry. Exogenous addition, contamination during processing, generation by endogenous microorganisms, and non-thermal plasma sterilization technology accumulate nitrite in fermented meat products, which is degraded by acids, enzymes, and other non-acid and non-enzyme substances. Nitrite levels are primarily affected by the matrix, fermentation conditions, and antioxidants in fermented meat products. The content of nitrite in fermented meat products can be reduced by inhibiting nitrite formation and substituting nitrite’s role in the fermentation process during meat processing. Future researchers should consider the following points: (i) The association between nitrite in plant extracts and nitrosamines based on their phenolic compounds has to be investigated using fermented meat products as the medium. (ii) More research is needed to evaluate the sensory acceptability of nitrite substitute products to the reformulated cured meat, taking into account consumer needs and concerns. (iii) Nitrite can be degraded by a variety of non-acid and non-enzyme compounds. Its composition and content, as well as the degradation effects of each component, and the synergistic degradation effects of each component and acid, should be analyzed further. (iv) Only CuNiRs have ever been explored in terms of their catalytic mechanism. Regarding the other three types of nitrite reductase (cd1NiRs, ccNiRs, and FdNiRs), their catalytic mechanisms have not been fully investigated. Excessive nitrite levels in cured meat products have become the principal issue influencing fermented meat product production. Further understanding of the nitrite metabolism in fermented meat products would support the appropriate application of nitrite in meat products and the production of more high-quality, low-nitrite fermented meat to meet consumer demand.

## Figures and Tables

**Figure 1 foods-12-01485-f001:**
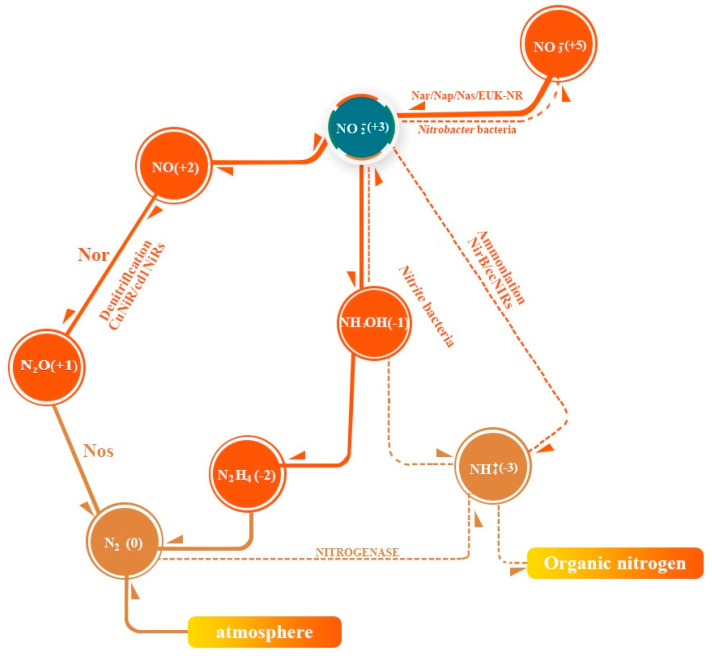
Biochemical cycling of nitrogen.

**Figure 2 foods-12-01485-f002:**
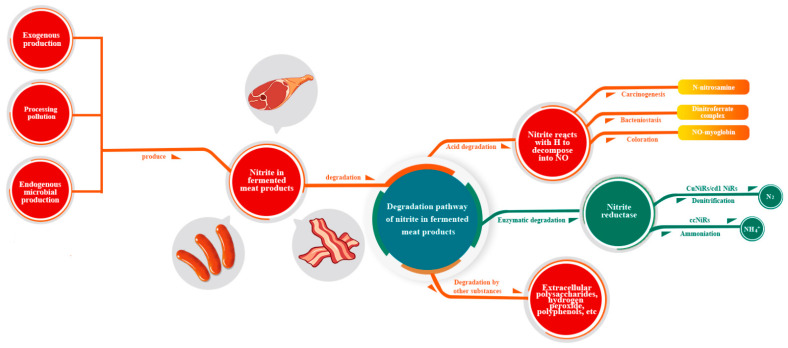
Metabolic pathway of nitrite in fermented meat products.

**Table 1 foods-12-01485-t001:** Application requirements of nitrite in different countries (only fermented meat products).

Country	Institution	Application Requirements of Nitrite	Reference
America	Code of Federal Regulations	The maximum permitted amount of nitrite use ≤ 200 mg/kg	[74]
Japan	Ministry of Health, Labor and Welfare of Japan	The maximum residue ≤ 70 mg/kg	[75]
Korea	Ministry of Food and Drug Safety	The maximum residue ≤ 70 mg/kg	[76]
Europe	European Commission	The maximum residue ≤ 50 mg/kg	[6]
China	State Health and Family Planning Commission	The maximum permitted amount ≤ 150 mg/kg (The maximum residue of sodium nitrite ≤ 30 mg/kg)	[77]

**Table 2 foods-12-01485-t002:** Studies on reducing the nitrite content in fermented meat products.

Methods		Products	Effects	Reference
Antioxidants substances	Sodium ascorbate (500 ppm) and α-Tocopherol acetate (10 ppm)	Fermented meat products	Reduced the content of nitrite, prevented the conversion of nitrite to nitrosamine, produced unique flavors, and improved the sensory quality.	[82]
	Ascorbic acid (0.0075%) and other curing additives	Beef sausage	Increased the antilisterial activity of *Enterococcus mundtii* to 2 log cfu/g.	[83]
	Phenolic	Fermented meat products	Phenolic compounds in bovine essential oils had antioxidant and antibacterial properties.	[84]
Lactic acid bacteria and its enzymes	*Limosilactobacillus fermentum* RC4 (1.06%) and *Lactiplantibacillus plantarum* B6 (0.53%)	No-nitrite-added cured meat	*Limosilactobacillus fermentum* RC4 had an effective nitrite degradation ability and *Lactiplantibacillus plantarum* B6 inhibited bacteria (0.53%).	[48]
	Enzymes	Pickle	Reduced nitrosamines.	[44]
Chromogenic agent	Beetroot and berries extracts	Fermented dry sausages	Imparted the red color of nitrite-free meat products to develop and stabilize color.	[85]
	Plasma-activated water	Beef jerky	Promoted the formation and fixation of color.	[86]
	Chili red and lycopene	Meat batters	Partially replaced nitrite from 150 mg/kg to 100 mg/kg, and improved the texture characteristics of the product.	[87]
Bacteriostat	Radish powder (0.5%) and chitosan (0.25%)	Fermented cooked sausages	Had good effects on physical, chemical and microbial stability of fermented cooked sausage.	[88]
	Garlic essential oil (125 mg/kg), allyl isothiocyanate (62.5 μL/kg), and nisin (20 mg/kg)	Fresh sausages	Effectively reduced *E. coli* O157H7 and LAB, and maintained the physical and chemical properties.	[89]

## Data Availability

The data presented in this study are available on request from the corresponding author.

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
