# Peer review of "Research Progress of Nitrite Metabolism in Fermented Meat Products"

_foods, 2023, doi:10.3390/foods12071485_

Round 1

Reviewer 1 Report

see attached file for the comments. 

Author Response

Response to Reviewer 1 Comments

Point 1:

Abstracts:

The full chemical name is needed to mention the first time.

Response 1Thank you for your suggestion. We have change “Lactobacillus” to “lactic acid bacteria" to make more appropriate in the manuscript. And we have revised “Lactobacillus” in the full text to make it more suitable for its original intention. We also checked other parts of the full text to ensure they are correct.

Point 2:

Introduction:

- At the end of the introduction, the authors may add how the authors collected all the

information to become this review. How many publications? Which year? The

database to obtain the papers?

Response 2Thank you for your suggestion. We added a bit of literature information.

Consequently, this review summarizes the research in the past five years, and discusses nitrite's source, degradation pathway, and conversion regulation from the four perspectives of nitrogen synthesis and degradation, nitrite metabolism, nitrite safety, and methods and application of fermented meat products, to provide a theoretical foundation for future research into nitrite and its application.

Point 3: 

Suggest changing topic No.2 (nitrogen synthesis and degradation in nature) to the

microbial related to nitrite metabolism. The low pH condition that affected the growth

of these bacteria should be added to understand the possibility of their growth during

the fermentation.

Response 3Thank you for your suggestion. We have changed topic No.2 (nitrogen synthesis and degradation in nature) to “Microorganisms related to nitrite metabolism”. And we add something about low pH condition that affected the growth of these bacteria.

Fermented meat products, which typically contain nitrite, a nitrogen-containing dietary additive, follow the same natural rhythm of nitrogen synthesis and breakdown. In general, the pH of meat products ranges between 5.5 and 6.2, while the pH of dry fermented sausage ranges between 4.5 and 5.5[14], both of which are optimum conditions(pH 4.5-5.5) that allow lactic acid bacteria to grow. However, N-nitrosamine formation may be more easily achieved in dry fermented sausages as the pH of the product approaches the optimum pH (pH 3.5) of the nitrosation reaction[14,15]. Nitrite is synthesized and degraded by bacteria in the endogenous system of fermented meat products, completing the nitrite metabolism cycle in these items[10].

Point 4: 

For topic 3.1 (Source of nitrite in fermented meat products), the nitrite from

exogenous addition contributes the most in fermented meat. Others are the very minor

source, and this information should add in this section.

Response 4Thank you for your suggestion. We add this information in 4.1 section.

Exogenous manufacture, contamination during processing, and endogenous microbial synthesis are the main sources of nitrite in fermented meat products. And a small amount of nitrite is produced by contamination during rearing, and endogenous microbial production. Acids, enzymes, and non-acid and non-enzyme compounds in bacteria can degrade nitrite in fermented meat products (Fig.2).

Point 5: 

Since nitrate is usually added in long-time ripening for fermented meat, what levels of nitrite and nitrate are typically added? How much are nitrate/nitrite concentrations in plant/plant extracts/ cultured plant extracts? These should add in section 3.1.1

Response 5Thank you for your suggestion. Through the different functions of nitrite, we added some contents about its levels of nitrite.

In meat products, nitrite serves four purposes. The first is the reaction of nitric oxide (NO) with hemoglobin and myoglobin to form nitrohemoglobin and nitrosymyoglobin, which maintain the bright color of meat products[16]. Jarulertwattana found that 4×10-6 g/g nitrite was suitable for curing chicken when used with ginger sauce, but more than 4×10-6 g/g nitrite will cause pink defects in chicken legs[17]. Second, nitrite can inhibit the growth of Clostridium botulinum and Staphylococcus aureus. The minimum concentration of nitrite that inhibits the outgrowth of Clostridium botulinum is from 4×10-5 to 8×10-5 g/g[1]. Third, nitrite has an antioxidant effect and can suppress lipid peroxidation, lowering the production of foul odors. Ji found that sodium nitrite can effectively inhibit lipid oxidation at 1×10-4 g/g during the curing process of mutton[18]. Finally, nitrite can aid some fermentation bacteria in the production of fermented flavor components in cured meat products through metabolism or the hydrolysis of protease and lipase.[4,19].

And we rewrote the content about vegetable extracts and added this content in section 3.1.1.

Some vegetable extracts are added to fermented meat products as nitrite substitutes because they contain a large amount of nitrite: celery, watercress, lettuce, spinach and rape (<2500 mg/100g); Chinese cabbage, leek, leek and parsley (1000 to <2500 mg/100g); radish, cabbage (500 to <1000 mg/100g); carrots, cucumbers, pumpkins and broccoli (200 to <500 mg/100g); potatoes, tomatoes, onions, eggplants, mushrooms and asparagus (<200 mg/100g)[20,21]. Meat products cured with vegetable extracts showed similar quality and sensory properties to nitrite cured products[22]. Natural nitrite sources have a higher consumer preference than synthetic nitrite because they contain a variety of antioxidants and antibacterial compounds. Phenolic compounds in vegetable extracts may hinder the conversion of nitrite into N-nitrosamines, making the probability of nitrosamines production lower than that of synthetic nitrite. However, there is no relevant research to prove its conversion relationship. Nevertheless, these plant extracts have many application limitations. Firstly, only vegetables which contain enough nitrate can be used. Furthermore, the taste and color of vegetables will also affect the sensory quality of fermented meat products. Besides, the transformation of nitrate to nitrite in plant extracts is uncontrollable, and the content of nitrite finally transformed may not reach the desired level[23].

Point 6:

Section 3.1.4, plasma sterilization could induce the formation of nitric oxide (not

nitrite). Thus, this should not include in topic 3.1 ""source of nitrite"". It suggested

moving to the end of the review to propose an alternative way to produce low-nitrite

fermented meat products.

Response 6Thanks, we have moved 3.1.4 section to 6.5 section.

Point 7:

The source of nitrite could be from the contamination of nitrate in water/ice and food

ingredients (such as sea salt) that can be chemically converted to nitrite through

microbial. This should be included.

Response 7Thank you for your suggestion. We believe that the source of nitrite may come from the pollution of nitrate in water/ice and food. It is more likely that nitrogen fertilizer is added to the environment, resulting in the imbalance of nitrogen cycle. We add this part to 3.1.2 Contamination during rearing

During animal rearing, domestic animals will ingest nitrite in the environment. Nitrogen compounds are widely used as fertilizers in agriculture. Crops produce plant chlorophyll in response to nitrogen fertilizer, which increases crop yield[24]. The use of nitrogen fertilizer has risen substantially in recent years, particularly in Southeast Asia[25]. Despite having a positive and significant impact on global food production, nitrogen fertilizer has a significant negative impact on the larger ecosystem. Under natural conditions, nitrifying microorganisms in the environment influence the nitrogen cycle of water, ensuring that nitrogen metabolism remains balanced. However, a large amount of high-nitrogen metabolites are added to aquaculture water, which outperforms the metabolism of natural bacteria in the reservoir, alters the dynamic balance of the nitrogen cycle, causes nitrite to deposit in the water, and eventually accumulates nitrite in animals[26].

Point 8:

Line 185-186. The authors mentioned that the acid and enzyme degradation occurred

utilizing Lactobacillus and denitrifying bacteria. Nitrite degradation to nitric oxide

could naturally occur by chemical reaction, especially at low pH. Thus, the

degradation of nitrite at various pH and Aw should be added in this review because it

is the primary condition of fermented meat products. The degree of degradation

occurs in which step the most (processing or storage)?

Response 8Thank you for your suggestion. We added a part about the degree of degradation.

In the curing process, lactic acid bacteria mostly focused in the processing step degrade nitrite rather than storage. According to Huang, Lactiplantibacillus fermentum RC4 and Lactiplantibacillus plantarum B6 were added to cured meat. The content of nitrite was significantly reduced (0.75 mg/kg) during processing, which was significantly lower than that of regular cured meat (4.67 mg/kg); During 0-20 days of storage, the moisture content of bacon decreased significantly, the pH increased continuously, but the content of nitrite did not change significantly[35].

Point 9:

Line 219-236 and Line 276-296 should not be here. It should be included in the topic

of nitrite toxicity/safety.

Response 9Thank you for your suggestion. We add a new setion 5.1 Toxicity of nitrite to include Line 219-236 and Line 276-296 in 5. Safety of nitrite in food.

Point 10:

Lines 238-275 should not be here, and they should include the role of nitrite in

fermented meat products.

Response 10Thank you for your suggestion. We add a new setion 3..The role of nitrite in fermented meat products to include Line 238-275.

Point 11:

The contents in topics 3.2.2 and 3.2.3 should be included in one.

Response 11Thank you for your suggestion. We deleted title of section 3.2.3. “Regulation of nitrite metabolism in microbes”. And we combined 3.2.2 and 3.2.3 in section 4.2.3. Other substances for degradation.

Point 12:

Contents in topics 3.2.4 and 3.3.3 should be combined. There is much information on using single and natural antioxidants to reduce the residual nitrite in meat products.

Thus, more information should be added here.

Response 12Thank you for your suggestion. We combined 3.2.4 and 3.3.3.

Point 13:

3.3.1 Sugar or polysaccharides that affect the degradation? Need more explanation

here. Simple Sugar or disaccharide? Hydrocolloid?

Response 13Thank you for your suggestion. We correct “sugar” to “Polysaccharides”, and explained the mechanism of it.

Polysaccharides and inorganic salts in the food matrix mainly affect the degradation of nitrite. The food matrix influences the activity of catalase in vitro of lactic acid bacteria and produces hydrogen peroxide, altering the ability of autolysis in vitro of lactic acid bacteria and influencing the activity of nitrate reductase in vitro[71]. 

Point 14:

Table 1, the table format makes it difficult to read. Table 1 suggests including only fermented meat products.

Response 14Thank you for your suggestion. We readjusted the table 1 to make it only include fermented meat products.

Table 1. Application requirements of nitrite in different countries(only fermented meat products).

Country

Institution

Application requirements of nitrite

Reference

Amercia

Code of Federal Regulations

The maximum permitted amount of nitrite use ≤ 200 mg/kg

[76]

Japan

Ministry of health, labor and welfare of Japan

The maximum residue ≤70mg/kg

[77]

Korea

Ministry of Food and Drug Safety

The maximum residue ≤70mg/kg

[78]

Europe

European Commission

The maximum residue ≤ 50 mg/kg

[6]

China

State health and Family Planning Commission

The maximum permitted amount ≤ 150 mg/kg

(The maximum residue of sodium nitrite ≤ 30 mg/kg)

[79]

Point 15:

Topic No.5 is the approach to producing low-nitrite fermented meat products, as the abstract mentions. More clarification needs here.

Response 15

Thank you for your suggestion. We rewrote the elaboration and abstract to discuss the approach to producing low-nitrite fermented meat products.

Recently, many studies have been conducted inhibiting of nitrite synthesis (Table 2). It is required to prevent the synthesis of nitrite in order to lower the level of nitrite in fermented products, such as by adding antioxidants and lactic acid bacteria, and replacing the role of nitrite with chromogenic agents and antibacterial agents. 

And we move 3.1.4. Non-thermal plasma sterilization to expand the methods and applications of substituting the nitrite in fermented meat products. In terms of content, we add main research contents in recent years to the table 2.

Reviewer 2 Report

Foods

foods-2265617

Research progress of nitrite metabolism in fermented meat products

Dear Editor,

The review article deals with the sources, degradation, limitations, and alteration techniques of nitrite. The topic of the ms is good. It has been generally well designed and written. However, it needs major revision. My specific comments and questions are below;

-       Line 5: And? One more researcher?

-       Lines 15, 19 and throughout the ms: “fermented meat” should be “fermented meat products”

-       Lines 33 and 37: Please mention their antioxidant properties as well.

-       Please mention that what is the novelty of your review compared to the researches in literature!

-       Lines 96 and 99 and throughout the ms: Please also mention the new names of these microorganisms such as Lactiplantibacillus plantarum!

-       Lines 117 and 122: Please give the minimum nitrite amounts for these effects!

-       Line 148: Please give the sources! What about spices used in preparation of meat products?

-       Lines 221 and 224: Please extend the disadvantage part of NO!

-       Line 381 and 382: Please give the mechanism!

Author Response

Response to Reviewer 2 Comments

Point 1:

-       Line 5: And? One more researcher?

Response 1Sorry for this, it is our negligence. We have deleted “And”.

Point 2:

-       Lines 15, 19 and throughout the ms: “fermented meat” should be “fermented meat products”

Response 2:Thank you for your suggestion. We have changed “fermented meat” to “fermented meat products”throughout the ms.

Point 3:

-       Lines 33 and 37: Please mention their antioxidant properties as well.

Response 3: Thank you for your suggestion. We have added antioxidant properties in Lines 33 and 37.

Fermented meat products were developed in response to a demand for meat storage. During the fermentation process, a series of biochemical and physical changes caused by microbial fermentation or enzymes provide the meat products with a unique flavor, color, texture, and antioxidant properties, so as to improve the edible quality of the meat[1]. Subsequently, nitrite, comprising salty white or light yellow particles[2], which primarily contain sodium or potassium nitrite, is commonly used in meat curing[3] as a color protectant, antioxidant and preservative, inhibiting the growth of spoilage and pathogenic bacteria, such as Clostridium botulinum and Listeria monocytogenes[4].

Point 4:

-       Please mention that what is the novelty of your review compared to the researches in literature!

Response 4:Thank you for your suggestion. What we want to point out is that acids, enzymes, and other compounds produced by lactic acid bacteria contribute to nitrite synthesis and degradation. We mentioned the novelty of our review at the end of introduction as below:

The lactic acid bacteria in fermented meat products are the key factors affecting nitrite metabolism, and the acids, enzymes, and other compounds produced by lactic acid bacteria contribute to nitrite synthesis and degradation.

Point 5:

-       Lines 96 and 99 and throughout the ms: Please also mention the new names of these microorganisms such as Lactiplantibacillus plantarum!

Response 5:

Thank you for your suggestion. Acorrding to Zheng JS et al. (2020) A  taxonomic  note  on  the  genus  Lactobacillus: Description of 23 novel genera, emended description of the genus Lactobacillus Beijerinck 1901, and union of Lactobacillaceae and Leuconostocaceae. Taxonomic Description template. https://doi.org/10.1099/ijsem.0.004107 

We have changed Lactobacillus plantarum to Lactiplantibacillus plantarum, Lactobacillus brevis to Levilactobacillus brevis, Lactobacillus curvatus to Latilactobacillus curvatus, Lactobacillus sakei to Latilactobacillus sakei, Lactobacillus rhamnosus to Lacticaseibacillus rhamnosus, Lactobacillus fermentum to  Limosilactobacillus fermentum.

Point 6:

-       Lines 117 and 122: Please give the minimum nitrite amounts for these effects!

Response 6:

Thank you for your suggestion. We added the minimum nitrite amounts for these effects. It was revised as follows:

The addition of nitrite to meat products during processing improves their appearance, color, flavor, and safety. It is also the main source of nitrite in fermented meat products. And a small amount of nitrite is produced by contamination during rearing, and endogenous microbial production. In meat products, nitrite serves four purposes. The first is the reaction of nitric oxide (NO) with hemoglobin and myoglobin to form nitrohemoglobin and nitrosymyoglobin, which maintain the bright color of meat products[16]. Jarulertwattana found that 4×10-6 g/g nitrite was suitable for curing chicken when used with ginger sauce, but more than 4×10-6 g/g nitrite will cause pink defects in chicken legs[17]. Second, nitrite can inhibit the growth of Clostridium botulinum and Staphylococcus aureus. The minimum concentration of nitrite that inhibits the outgrowth of Clostridium botulinum is from 4×10-5 to 8×10-5 g/g[1]. Third, nitrite has an antioxidant effect and can suppress lipid peroxidation, lowering the production of foul odors. Ji found that sodium nitrite can effectively inhibit lipid oxidation at 1×10-4 g/g during the curing process of mutton[18]. Finally, nitrite can aid some fermentation bacteria in the production of fermented flavor components in cured meat products through metabolism or the hydrolysis of protease and lipase.[4,19].

Point 7:

-       Line 148: Please give the sources! What about spices used in preparation of meat products?

Response 7:Thank you for your suggestion. We added sources about it. It was revised as follows:

During animal rearing, domestic animals will ingest nitrite in the environment. Nitrogen compounds are widely used as fertilizers in agriculture. Crops produce plant chlorophyll in response to nitrogen fertilizer, which increases crop yield[24]. The use of nitrogen fertilizer has risen substantially in recent years, particularly in Southeast Asia[25]. Despite having a positive and significant impact on global food production, nitrogen fertilizer has a significant negative impact on the larger ecosystem. Under natural conditions, nitrifying microorganisms in the environment influence the nitrogen cycle of water, ensuring that nitrogen metabolism remains balanced. However, a large amount of high-nitrogen metabolites are added to aquaculture water, which outperforms the metabolism of natural bacteria in the reservoir, alters the dynamic balance of the nitrogen cycle, causes nitrite to deposit in the water, and eventually accumulates nitrite in animals[26].

This part mainly describes the accumulation of a small amount of nitrite in animals through external environment. Through literature review, we believe that spices should not be one of the main sources of nitrite. On the contrary, some spices can play a role in degrading nitrite, such as clove. We mentioned it in section 5.1 Antioxidant substitutes section.

Some natural ingredients not only have an antioxidant effect, but also have a potential inhibitory effect on controlling microbiological hazards, such as cloves, essential oils of aromatic and medicinal plants, plant extracts with high polyphenols concentration, acidified whey, honey, and other bee products[21].

Point 8:

-       Lines 221 and 224: Please extend the disadvantage part of NO!

Response 8:Thank you for your suggestion. We added the disadvantage part of NO. It was revised as follows:

Nitrite (NO2-) forms nitrite acid (HNO2) when it combines with hydrogen ions (H+). Nitric acid then gradually decomposes into nitrous trioxide (N2O3) and water molecules (H2O) (Equal. 6 and 7). N2O3 further dissociates into NO and NO2 (Equal. 8). NO can further form N-nitrosamines with carcinogenic and teratogenic effects[3].

As for carcinogenicity of NO, we have set a new chapter 5.1. Toxicity of nitrite, which included the disadvantage part of NO.

NO is a crucial compound generated from nitrite, which combines with myoglobin and methemyoglobin, proteins in cells, and with amines to lower NO levels in the dynamic balance, resulting in nitrite degradation. In meat fermentation, NO has coloring, antioxidant, cured flavor, and antibacterial benefits; however, it can also be carcinogenic. At low concentrations (nm), NO acts as a signaling molecule for intercellular communication in neurons and the cardiovascular system[70]. At high concentrations (μM) NO can kill pathogens and cancer cells[71].

Through a nucleophilic substitution reaction, NO can combine with non-protonated secondary amines to form N-nitrosamines, which are carcinogenic and teratogenic[3]. The most common volatile N-nitrosamines in meat products are N-nitrosodimethylamine, N-nitrosodiethylamine, N-nitrosopiperidine, N-nitrosopyrrolidine, and N-nitrobenzylmorpholine. Among them, in terms of oncogenicity and genotoxicity, N-nitrosodimethylamine and N-nitrosodiethylamine are regarded as the most volatile N-nitrosamines[72]. The amount of N-nitrosamines production mainly depends on the intake of nitrite and the processing conditions in meat products, which also increases in concentration under conditions of pH 2.5~3.5, longer storage time, high temperature and high acid conditions[73]. The presence of amines is one of the primary prerequisites for the creation of N-nitrosamines. There aren't many N-nitrosamines in raw meat since amino acids are only decarboxylated to produce amine. Nevertheless, procedures including maturation, fermentation, and curing may increase their synthesis of N-nitrosamines. It was discovered during processing that the quantity of precursors was directly correlated with the amount of N-nitrosamines in meat products, but even if precursors were present, low water activity and an unfavorable pH would prevent the production of nitrosamines in meat products [74]. Additionally, several cooking techniques that reach temperatures greater than 130 °C, like frying or barbecuing, may make it more likely that N-nitrosamines will occur[3]. To prevent nitrosation, chemicals like ascorbate and tocopherol are added during processing. The presence of sulfhydryl compounds, some phenols, and tannins in meat products might also inhibit the formation of N-nitrosamines[15].

Point 9:

-       Line 381 and 382: Please give the mechanism!

Response 9:Thank you for your suggestion. We added content about mechanism. It was revised as follows:

Polysaccharides and inorganic salts in the food matrix mainly affect the degradation of nitrite. The food matrix influences the activity of catalase in vitro of lactic acid bacteria and produces hydrogen peroxide, altering the ability of autolysis in vitro of lactic acid bacteria and influencing the activity of nitrate reductase in vitro[71]. Seo et al studied the extracellular polysaccharides produced by Lactiplantibacillus plantarum YML009[72]. The nitrite clearance rate was 43.93%, which proved the extracellular polysaccharides have a strong ability to degrade nitrite, and the higher the sugar content, the higher the nitrite-free radical clearance rate. Inorganic salt can reduce the water content of fermented meat products and inhibit the growth of microorganisms. When excessive sodium chloride (NaCl) is used to pickle fermented meat products, bacteria will dehydrate and die, and the nitrite degradation effect will be significantly reduced. Delgado-Pando et al evaluated bacon with different salt contents and found that the color of bacon with a high salt content was redder than that with low salt content[73].
